# Association between lifestyle-related factors and low back pain: Evidence from a Japanese population–based study

Soya Kawabata[1,2], Noriaki Kurita[2,3], Takuya Nikaido[2,4], Ryoji Tominaga[2,3,5], Yuji Endo[2,4], Nobuyuki Fujita[1,2]*, Shin-ichi Konno[2,4], Seiji Ohtori[2,6]

1 Department of Orthopaedic Surgery, Fujita Health University, School of Medicine, Toyoake, Aichi, Japan, 2 Clinical Research Committee of the Japanese Society of Lumbar Spine Disorders, Fukushima, Japan, 3 Department of Clinical Epidemiology, Graduate School of Medicine, Fukushima Medical University, Fukushima, Japan, 4 Department of Orthopaedic Surgery, Fukushima Medical University School of Medicine, Fukushima, Japan, 5 Department of Orthopaedic Surgery, Iwai Orthopaedic Hospital, Edogawa-ku, Tokyo, Japan, 6 Department of Orthopaedic Surgery, Graduate School of Medicine, Chiba University, Chiba, Japan

* nfujita2007@gmail.com

## Abstract

Low back pain (LBP) is a major public health issue, and lifestyle-related factors (LRFs) are increasingly recognized as key contributors to LBP. However, comprehensive studies using recent data concerning the association between LBP and LRFs remain limited. In this study, a nationally representative sample of Japanese adults were surveyed to evaluate the relationship between LRFs and LBP and to explore how these factors relate to both the severity and chronicity of LBP. A cross-sectional nationwide survey was conducted among 5000 randomly selected Japanese adults aged 20–90 years; valid responses were obtained from 2188. Participants were analyzed using three different methods: (1) those with or without current LBP, (2) those with no/mild or moderate/severe pain, and (3) those with or without chronic LBP. Key LRFs included body mass index, alcohol consumption, smoking, exercise habits, comorbidities (dyslipidemia, diabetes, and hypertension), and self-image regarding body shape. Multivariable logistic regression analysis revealed that current LBP was significantly associated with body mass index (odds ratio [OR]=1.04, 95% confidence interval [CI]: 1.00–1.07), alcohol consumption (OR=1.37, 95% CI: 1.04–1.80), smoking (OR=1.63, 95% CI: 1.21–2.20), and dyslipidemia (OR=1.51, 95% CI: 1.06–2.13), and the severity of LBP was associated with smoking (OR=1.77, 95% CI: 1.19–2.64), lack of exercise (OR=1.55, 95% CI: 1.10–2.15), and dyslipidemia (OR=1.64, 95% CI: 1.06–2.55). In addition, smoking was the only LRF significantly associated with chronic LBP (OR=1.70, 95% CI: 1.23–2.34). Multiple LRFs are associated with the prevalence of LBP. Stratified analysis provided deeper insight into specific risk factors for LBP. In particular, dyslipidemia is linked to pain severity, whereas smoking is associated with both severity and chronicity. Future longitudinal studies should focus on the influence of these key LRFs on onset, severity, and chronicity of LBP.

**Data availability statement:** With regard to data availability, the full dataset cannot be made publicly available due to ethical restrictions and participant confidentiality, as informed consent for open data sharing was not obtained and the data contain potentially identifying information. Interested researchers may request access to the data by contacting the Ethics Committee of Fukushima Medical University (chiken@fmu.ac.jp), pending approval from the Ethics Committee.

**Funding:** This study was supported by a scholarship donation from Eli Lilly Japan K.K., and a research grant from the Japanese Society of Lumbar Spine Disorders. The funders had no role in study design, data collection and analysis, decision to publish, or preparation of the manuscript.

**Competing interests:** This study was supported by a scholarship donation from Eli Lilly Japan K.K. This does not alter our adherence to PLOS ONE policies on sharing data and materials.

## Introduction

Low back pain (LBP) is a very common symptom in all age groups around the world. From 1990 to 2015, the number of individuals experiencing daily life disabilities caused by LBP increased by 54% [1]. Approximately 70%–85% of all people experience LBP at least once in their lifetime [2]. LBP imposes a substantial societal burden beyond medical expenses, including labor loss and increased healthcare utilization, which ultimately affects national economies [2,3]. Beyond its economic and occupational consequences, LBP profoundly impairs patients' psychological and social well-being. A study of the psychosocial distress of patients with chronic LBP revealed that difficulties in daily activities caused severe frustration in affected patients. However, because many such patients appear physically healthy, the cause of their suffering is not apparent. Consequently, their pain is frequently dismissed as a psychological issue, which further worsens their distress [3]. These findings underscore the urgent need for global efforts to reduce the prevalence of LBP.

In recent years, adverse lifestyle factors such as physical inactivity, obesity, smoking, and poor diet have been recognized as risk factors for LBP, and maintaining a healthy lifestyle is considered important for both the prevention and mitigation of symptoms [4–7]. Similarly, lifestyle-related diseases such as diabetes, hypertension, and dyslipidemia—which are closely associated with these behaviors—have also been identified as potential contributors to LBP [8–10]. Beyond these physical and clinical indicators, self-image regarding body shape—a form of subjective health perception—has also gained attention as a psychosocial factor influencing lifestyle-related diseases and overall health status [11]. Although numerous studies have focused on the relationship between LBP and either lifestyle factors or lifestyle-related diseases, few have focused on both associations together with regard to the broader concept of lifestyle-related factors (LRFs) [4]. Moreover, LBP varies not only in its intensity but also in its duration. However, to the best of our knowledge, the associations between LRFs and both the severity and chronicity of LBP have not been explored simultaneously.

In 2023, the Clinical Research Committee of the Japanese Society of Lumbar Spine Disorders conducted a nationwide survey to better understand the social and health-related impact of LBP in contemporary Japan [4]. Using data from this large-scale survey, the present study aimed to comprehensively examine the relationship between LRFs and LBP in the Japanese general population. Specifically, we investigated whether LRFs are associated with both the severity and chronicity of LBP, with the ultimate goal was of identifying key modifiable factors to guide future public health strategies for prevention and management.

## Materials and methods

### Study overview

We used the same target population, research design, and protocol as in the previous primary survey [4].

## Study participants, design, and protocol

For this cross-sectional observational study, approximately 5000 Japanese adults were randomly selected to receive a survey about LBP. The survey was carried out between June 17 and July 17, 2023, using a self-administered placement survey method.

A stratified two-stage random sampling method was employed, dividing Japan into 43 strata based on regions and city sizes. A total of 250 locations were randomly selected on the basis of the population distribution according to the 2020 census. Twenty individuals from each location were randomly chosen for this study; thus, the initial sample size was 5000. Before the survey, a letter requesting participation was sent in advance to the selected individuals. Subsequently, trained surveyors visited their homes and hand-delivered the self-administered survey forms enclosed in detention envelopes. Upon retrieval, the surveyors confirmed directly with the participants or their household members that the forms had been completed by the participants themselves. To ensure accuracy, 20% of the respondents received double postcards (i.e., prepaid return postcards) to confirm the proper distribution and collection of the survey forms. Participants were rewarded with a gift card for completing the survey. The sampling, data collection, and data entry were conducted at the Nippon Research Center, Ltd., Tokyo, Japan.

## Ethics approval

This study was approved by the ethics committees of Fukushima Medical University (approval number: REC2022−024). Written informed consent was obtained from all participants. The study adhered to the guidelines of the Declaration of Helsinki.

## Definition of LBP

LBP was defined as pain localized to the posterior region of the body, specifically between the 12th rib and the lower gluteal folds, that persisted for at least 24 h. To ensure clarity in identifying the affected area, participants were given an illustration of a human figure, with the region between the 12th rib and the lower gluteal folds clearly marked. Participants who answered "yes" to the question "Have you experienced LBP lasting more than 24 hours in the past month?" were classified as having current LBP. Among them, those who responded that their pain began more than 3 months earlier were classified as having chronic LBP.

## Measured items

A self-administered questionnaire was used to investigate various factors related to LBP and its impact on participants. The main evaluation items included the presence of LBP, duration of LBP, severity of LBP (assessed on a visual analogue scale [VAS]), age, sex, body mass index (BMI), comorbidities (dyslipidemia, diabetes, and hypertension); however, treatment status for these comorbidities was not assessed, smoking and drinking habits, exercise habits (equivalent to walking for at least 1 hour per day), and current self-image regarding body shape. Participants were considered smokers if they had smoked in the past month, and participants who consumed alcohol at least three times per week were considered drinkers. For self-image assessment, participants were shown nine gender-specific silhouettes and were asked to select the one that best represented their current body shape (Fig 1). Silhouettes 1–3 represented "underweight," 4–6 represented "normal weight," and 7–9 represented "obese." These silhouette figures were based on a previous study on body image perception in Japanese and Vietnamese adolescents [5].

## Evaluations

The following three descriptions were used to analyze various evaluation items: (1) the presence or absence of current LBP (i.e., LBP experienced within the past month); (2) current severity of LBP according to the VAS score, categorized

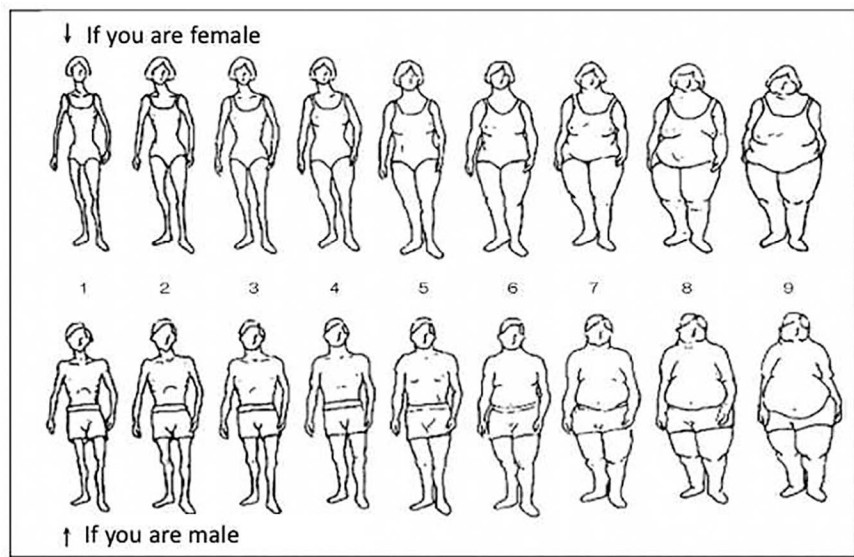

**Fig 1. Gender-specific silhouettes for self-image assessment.** Participants were shown nine silhouettes and asked to select the one that best represented their current body shapes. The selected number was used to categorize body shape into three groups: 1–3 as "underweight," 4–6 as "normal weight," and 7–9 as "obese".

into two groups; and (3) the presence or absence of chronic LBP. In accordance with previous reports [6], VAS scores of less than <50 mm represented "No/mild pain," and scores of 50 mm or more represented "moderate/severe pain." In addition, chronic LBP was defined as pain persisting for 3 months or longer [7].

## Statistical analysis

We used two-tailed *t* tests and Pearson's chi-square tests for group comparisons, and we performed multivariable analysis with multiple logistic regression. Missing values in explanatory variables were handled by listwise deletion. The proportion of excluded cases was small and missingness was assumed to be at random. Covariates for the multiple logistic regression models were selected according to biological plausibility and prior literature [8–10]. Specifically, different sets of variables previously associated with LBP were forced into each model. To perform statistical analyses, we used IBM SPSS Statistics Version 29.0 (IBM Corp., Armonk, NY, USA); *P* values of <0.05 indicated statistical significance.

## Results

Of the 5000 eligible participants in this study, 2188 (43.8%) returned the survey in the primary study [4], and we analyzed their responses in this study. Participants' mean age was 56.0 years, and 47.1% were men.

### Comparison between participants with and without current LBP

After excluding four participants who provided unclear responses regarding the presence of LBP, the group of respondents without LBP in the previous month consisted of 1856 participants; the group of those who had experienced LBP in the previous month (current LBP) included 328 participants (Table 1). The multivariable logistic regression analysis revealed that several factors were significantly associated with current LBP (Table 2). The mean age of the participants without LBP was 55.4±0.4 years, whereas those with current LBP were significantly older (59.0±0.9 years; odds ratio [OR]=1.012, 95% confidence interval [CI]: 1.004–1.02, *P*=0.004). The mean BMI of the participants with current LBP (23.4±0.2) was significantly higher than that of participants without LBP (22.8±0.1; OR=1.036, 95% CI: 1.002–1.072,

**Table 1. Comparison between participants with and without current LBP.**

| | | Non-current LBP group (n=1856) | | Current LBP group (n=328) | | p value |
|---|---|---|---|---|---|---|
| Sex | | Men | 872 | Men | 158 | 0.849 |
| | | Women | 983 | Women | 170 | |
| Age (yeas) | | 55.4±0.4 | | 59.0±0.9 | | <0.001 |
| BMI (kg/m2) | | 22.8±0.1 | | 23.4±0.2 | | 0.004 |
| Alcohol consumption (≥3 times per week) | | 480/1847 (26.0%) | | 111/328 (33.8%) | | 0.003 |
| Smoking in the past month | | 330/1846 (17.9%) | | 83/328 (25.3%) | | 0.002 |
| Lack of regular exercise (equivalent to walking ≥1 hour/day) | | 742/1846 (40.2%) | | 147/326 (45.1%) | | 0.156 |
| Comorbidities | Dyslipidemia | 191/1856 (10.3%) | | 53/328 (16.2%) | | 0.002 |
| | Diabetes | 122/1856 (6.6%) | | 27/328 (8.2%) | | 0.272 |
| | Hypertension | 446/1856 (24.0%) | | 103/328 (31.4%) | | 0.005 |
| Current self-image | | Underweight | 483 (26.0%) | Underweight | 83 (25.3%) | 0.077 |
| | | Normal weight | 1223 (65.9%) | Normal weight | 206 (62.8%) | |
| | | Obese | 150 (8.1%) | Obese | 39 (11.9%) | |

T-test or Chi-square test.

**Table 2. Multivariable Logistic Regression Analysis of Risk Factors for Current LBP.**

| | | OR | 95% CI | | p value |
|---|---|---|---|---|---|
| Age (yeas) | | 1.012 | 1.004 | 1.02 | 0.004 |
| BMI (kg/m2) | | 1.036 | 1.002 | 1.072 | 0.039 |
| Alcohol consumption (≥3 times per week) | | 1.366 | 1.039 | 1.796 | 0.025 |
| Smoking in the past month | | 1.628 | 1.205 | 2.198 | 0.001 |
| Lack of regular exercise (equivalent to walking ≥1 hour/day) | | 0.803 | 0.63 | 1.024 | 0.077 |
| Comorbidities | Dyslipidemia | 1.505 | 1.064 | 2.129 | 0.021 |
| | Diabetes | 0.966 | 0.614 | 1.52 | 0.881 |
| | Hypertension | 1.048 | 0.779 | 1.409 | 0.757 |

OR, odds ratio; CI, confidence interval.

$P=0.039$). With regard to lifestyle habits, the proportion of participants who consumed alcohol three or more times per week was significantly higher among those with current LBP (33.8%) than among those without LBP (26.0%; OR=1.366, 95% CI: 1.039–1.796, $P=0.025$). In addition, the proportion of participants who had smoked in the past month was significantly higher among those with current LBP (25.3%) than among those without LBP (17.9%; OR=1.628, 95% CI: 1.205–2.198, $P=0.001$). With regard to comorbidities, dyslipidemia and hypertension were significantly more prevalent among participants with current LBP than among those without LBP; the presence of dyslipidemia, in particular, was significantly associated with current LBP (OR=1.505, 95% CI: 1.064–2.129, P=0.021).

## Comparison based on current LBP severity

Next, participants were classified into two groups on the basis of their VAS scores for current LBP: those with no/mild LBP (n=2019) and those with moderate/severe LBP (n=158; Table 3). The multivariate logistic regression analysis identified several factors significantly associated with the severity of LBP (Table 4). Those with moderate/severe pain were, on average, older (60.9±1.3 years) than those with no/mild pain (55.5±0.4 years; OR=1.019, 95% CI: 1.007–1.03, $P=0.001$).

**Table 3. Comparison Based on current LBP Severity.**

| | | VAS | | | | | p value |
|---|---|---|---|---|---|---|---|
| | | No/mild pain group (n = 2019) | | | Moderate/severe pain group (n = 158) | | |
| Sex | | Men | 954 | | Men | 71 | 0.819 |
| | | Women | 1064 | | Women | 87 | |
| Age (yeas) | | 55.5 ± 0.4 | | | 60.9 ± 1.3 | | <0.001 |
| BMI (kg/m2) | | 22.8 ± 0.1 | | | 23.6 ± 0.3 | | 0.020 |
| Alcohol consumption (≥3 times per week) | | 544/2019 (26.9%) | | | 45/158 (28.5%) | | 0.652 |
| Smoking in the past month | | 371/2019 (18.4%) | | | 41/158 (25.9%) | | 0.047 |
| Lack of regular exercise (equivalent to walking ≥1 hour/day) | | 804/2007 (40.1%) | | | 81/158 (51.3%) | | 0.006 |
| Comorbidities | Dyslipidemia | 214/2019 (10.6%) | | | 30/158 (19.0%) | | 0.001 |
| | Diabetes | 132/2019 (6.5%) | | | 17/158 (10.8%) | | 0.043 |
| | Hypertension | 492/2019 (24.4%) | | | 53/158 (33.5%) | | 0.010 |
| Current self-image | | Underweight | 524 (26.0%) | | Underweight | 40 (25.3%) | 0.164 |
| | | Normal weight | 1328 (65.8%) | | Normal weight | 98 (62.0%) | |
| | | Obese | 167 (8.3%) | | Obese | 20 (12.7%) | |

T-test or Chi-square test.

**Table 4. Multivariable Logistic Regression Analysis of Risk Factors for Severe Pain in Current LBP.**

| | | OR | 95% CI | | | p value |
|---|---|---|---|---|---|---|
| Age (yeas) | | 1.019 | 1.007 | | 1.03 | 0.001 |
| BMI (kg/m2) | | 1.037 | 0.991 | | 1.084 | 0.118 |
| Alcohol consumption (≥3 times per week) | | 0.975 | 0.669 | | 1.42 | 0.893 |
| Smoking in the past month | | 1.773 | 1.192 | | 2.636 | 0.005 |
| Lack of regular exercise (equivalent to walking ≥1 hour/day) | | 1.546 | 1.11 | | 2.153 | 0.010 |
| Comorbidities | Dyslipidemia | 1.641 | 1.055 | | 2.554 | 0.028 |
| | Diabetes | 1.15 | 0.659 | | 2.006 | 0.623 |
| | Hypertension | 1.049 | 0.706 | | 1.558 | 0.814 |

OR, odds ratio; CI, confidence interval.

Similarly, those with moderate/severe pain had a significantly higher mean BMI (23.6 ± 0.3) than did those with no/mild pain (22.8 ± 0.1). With regard to lifestyle habits, the proportion of participants who had smoked in the past month was significantly higher among those with moderate/severe pain (25.9%) than among those with no/mild pain (18.4%; OR=1.773, 95% CI: 1.192–2.636, *P*=0.005). In addition, the proportion of participants without a habit of walking was significantly higher among those with moderate/severe pain (51.3%) than among those with no/mild pain (40.1%; OR=1.546, 95% CI: 1.11–2.153, *P*=0.01). With regard to comorbidities, dyslipidemia, diabetes, and hypertension were all significantly more prevalent among participants with moderate/severe pain (for dyslipidemia, OR=1.641, 95% CI: 1.055–2.554, *P*=0.028).

## Comparison between participants with and without chronic LBP

Participants were further classified according to the presence or absence of chronic LBP (Table 5): 1931 participants had non-chronic LBP, and 250 had chronic LBP. Multivariable logistic regression analysis revealed that the factors significantly

**Table 5. Comparison between participants with and without chronic LBP.**

| | | Non-chronic LBP group (n = 1931) | | Chronic LBP group (n = 250) | | p value |
|---|---|---|---|---|---|---|
| Sex | | Men | 914 | Men | 115 | 0.864 |
| | | Women | 1016 | Women | 135 | |
| Age (yeas) | | 55.4 ± 0.4 | | 60.4 ± 1.1 | | < 0.001 |
| BMI (kg/m2) | | 22.8 ± 0.1 | | 23.5 ± 0.3 | | 0.008 |
| Alcohol consumption (≥3 times per week) | | 511/1922 (26.6%) | | 79/250 (31.6%) | | 0.094 |
| Smoking in the past month | | 349/1921 (18.2%) | | 63/250 (25.2%) | | 0.008 |
| Lack of regular exercise (equivalent to walking ≥1 hour/day) | | 779/1920 (40.6%) | | 110/249 (44.2%) | | 0.277 |
| Comorbidities | Dyslipidemia | 205/1931 (10.6%) | | 39/250 (15.6%) | | 0.019 |
| | Diabetes | 124/1931 (6.4%) | | 25/250 (10.0%) | | 0.035 |
| | Hypertension | 466/1931 (24.1%) | | 82/250 (32.8%) | | 0.003 |
| Current self-image | | Underweight | 503 (26.0%) | Underweight | 63 (25.2%) | 0.025 |
| | | Normal weight | 1272 (65.9%) | Normal weight | 154 (61.6%) | |
| | | Obese | 156 (8.1%) | Obese | 33 (13.2%) | |

T-test or Chi-square test.

associated with chronic LBP were age and smoking (Table 6). The participants with chronic LBP were, on average, significantly older (60.4 ± 1.1 years) than those with non-chronic LBP (55.5 ± 0.4 years; OR=1.017, 95% CI: 1.008–1.027, $P < 0.001$). Similarly, participants with chronic LBP had a significantly higher mean BMI (23.5 ± 0.3) than did those with non-chronic LBP (22.8 ± 0.1). Regarding lifestyle habits, the proportion of participants who smoked was significantly higher among those with chronic LBP (25.2%) than among those with non-chronic LBP (18.2%; OR=1.695, 95% CI: 1.229–2.337, $P = 0.001$). In addition, with regard to comorbidities, dyslipidemia, diabetes, and hypertension were all significantly more prevalent among participants with chronic LBP group. In terms of current self-image, the proportion of participants who perceived themselves as "obese" was higher among those with chronic LBP than among those with non-chronic LBP.

## Discussion

We examined the association between LBP and LRFs in a nationally representative sample from Japan. A key feature of this study was the classification of participants according to the presence or absence of LBP in the past month, pain severity (VAS score), and the presence or absence of chronic LBP. This approach allowed us to identify not only factors associated with the presence or absence of LBP but also those related to pain intensity and duration; these findings provide a more comprehensive understanding of the condition.

**Table 6. Multivariable Logistic Regression Analysis of Risk Factors for chronic LBP.**

| | | OR | 95% CI | | p value |
|---|---|---|---|---|---|
| Age (yeas) | | 1.017 | 1.008 | 1.027 | < 0.001 |
| BMI (kg/m2) | | 1.031 | 0.985 | 1.079 | 0.193 |
| Smoking in the past month | | 1.695 | 1.229 | 2.337 | 0.001 |
| Comorbidities | Dyslipidemia | 1.298 | 0.883 | 1.909 | 0.185 |
| | Diabetes | 1.191 | 0.746 | 1.903 | 0.464 |
| | Hypertension | 1.062 | 0.768 | 1.469 | 0.716 |
| Current self-image | | 1.071 | 0.793 | 1.447 | 0.656 |

Our multivariable logistic regression analysis identified older age, high BMI, alcohol consumption, smoking, and dyslipidemia as independent risk factors for current LBP. Furthermore, smoking, lack of exercise, and dyslipidemia were significantly associated with pain severity, and older age and smoking were independently associated with chronic LBP. These findings suggest that LRFs play a crucial role in LBP occurrence, intensity, and chronicity. Of note is that although diabetes and hypertension were initially associated with LBP in the unadjusted analyses, they were not significant in multivariable models, which indicates that their effects may be confounded by other factors. Meanwhile, dyslipidemia was significantly associated with both the occurrence and severity of LBP. This result is consistent with those of previously reported studies. In a study involving 1035 participants in China, Yuan et al. demonstrated that dyslipidemia is a risk factor for lumbar intervertebral disc degeneration and Modic changes [12]. They reported that high levels of both total cholesterol and low-density lipoprotein C (LDL-C) were independent risk factors for disc degeneration, which is consistent with our findings. Heuch et al. found that serum lipid abnormalities are associated with chronic LBP and that higher LDL-C levels may potentially increase the risk of LBP [13]. Furthermore, Yoshimoto et al. suggested that impaired blood flow to the lumbar spine as a result of atherosclerosis may contribute to LBP. Their findings particularly emphasized the association between an increased ratio of LDL-C levels to high-density lipoprotein C levels and LBP risk [14]. These findings imply that dyslipidemia may induce endothelial dysfunction, impair nutrient supply to intervertebral discs, and consequently accelerate disc degeneration. In addition to disc degeneration, previous research has suggested that dyslipidemia may be involved in other lumbar pathologies such as spinal epidural lipomatosis and canal stenosis, which could contribute to both the presence and severity of LBP [15]. Moreover, a study conducted among women revealed that dyslipidemia was significantly associated with risk for chronic LBP [16]. Our results showed that although dyslipidemia was not significantly associated with chronic LBP, it may contribute to the severity of LBP, as well as to its presence. These discrepancies may be explained by confounding factors such as obesity and physical inactivity, which are often associated with both dyslipidemia and chronic LBP. For example, in a large cohort study involving over 18,000 adults, low HDL-C and high triglyceride levels were initially associated with an increased risk of chronic LBP. However, these associations disappeared after adjusting for body mass index and lifestyle variables [8]. This suggests that the link between dyslipidemia and chronic pain may be mediated by adiposity rather than a direct causal effect. In our study, the absence of a significant association between dyslipidemia and chronic LBP may similarly reflect the influence of unmeasured confounders or mediating variables.

According to our findings, smoking was significantly associated with both current and chronic LBP. Moreover, smoking was significantly related to pain severity. Among the three analyses conducted in this study, smoking was the only LRF that showed such significant associations. This finding suggests that smoking may be the LRF with the strongest effects on LBP. Similarly, a meta-analysis by Shiri et al. indicated that smoking increases the risk of current, chronic, and severe LBP [17], which supports our findings. Furthermore, Vaajala et al. showed that adolescent smoking is associated with an increased risk of developing LBP in adulthood, as well as a higher likelihood of LBP-related hospitalization and spinal surgery [18]. The potential mechanisms through which smoking contributes to LBP include reduced blood flow to the lumbar region, which impairs nutrient supply to the intervertebral discs and thereby accelerates disc degeneration. In addition, smoking has been reported to increase levels of inflammatory cytokines, which may amplify chronic pain. The results of this study suggest that smoking may contribute to the chronicity of LBP, which highlights the importance of smoking prevention and cessation support as part of chronic LBP management. Furthermore, this study identified alcohol consumption as an independent risk factor for current LBP. According to a systematic review by Ferreira et al., alcohol consumption is slightly associated with LBP; however, no dose-response relationship was identified [19]. Additionally, Vaajala et al. reported that excessive alcohol intake during adolescence increases the risk of developing LBP in adulthood [18]. Based on these findings, it is advisable to avoid excessive alcohol consumption from the perspective of LBP prevention.

In this study, habitual exercise equivalent to 1 hour of walking per day was not significantly associated with the presence of either current or chronic LBP. However, the results indicated that a lack of exercise habits was associated with

increased pain severity. Exercise may therefore contribute to pain relief, but it is also possible that severe pain may prevent patients from exercising; thus, determining the causality of exercise is challenging. Previous studies have shown that physical inactivity is a risk factor for LBP and that exercise may help prevent its onset [20].

Obesity has been widely reported as a risk factor for LBP [21,22]. Our findings indicated that high BMI is an independent risk factor for current LBP. However, in terms of self-image, there was no significant difference between the groups except for participants with chronic LBP, which suggests that many participants may have underestimated body weight. According to Duncan et al., a significant proportion of adults in the United States—71% of men and 65% of women—misperceive their body weight despite being overweight or obese [23]. The study emphasized that correcting this misperception is crucial for raising awareness of weight management and promoting weight loss behaviors. Meanwhile, some authors have pointed out that BMI may not always be a reliable indicator of obesity; thus, BMI alone as a measure should be used cautiously [24]. Nevertheless, fostering accurate self-assessment and promoting lifestyle improvements through education may play a key role in preventing LBP.

A major strength of this study was its relatively large-scale survey targeting the Japanese general population. This comprehensive approach allowed for a more extensive evaluation of the association between LBP and LRFs. However, this study had several limitations. First, as a cross-sectional study, it could not establish causality. For instance, it remains unclear whether dyslipidemia contributes to the development of LBP or whether LBP-related reduction in physical activity leads to dyslipidemia. To clarify these relationships, further longitudinal studies are needed. Second, the diagnosis of LBP in this study relied on self-reported data without the use of objective assessment measures. This limitation should be taken into consideration in interpreting the findings. Third, the rate of response to the mail survey was low (43.8%), which may have influenced the results. However, it is noteworthy that the response rates have declined in recent years even in large-scale national surveys, such as the U.S. National Health Interview Survey (NHIS). For instance, during the coronavirus disease 2019 (COVID-19) pandemic, NHIS reported a similar response rate of approximately 42%, which highlights the broader challenges of survey participation under current conditions [25]. Despite these limitations, our study provides valuable insights into the epidemiologic characteristics of LBP in the Japanese population.

In conclusion, we identified dyslipidemia and smoking as key LRFs that are associated with LBP. Future longitudinal studies are needed to examine whether these key LRFs influence the onset, severity, and chronicity of LBP.

## Author contributions

**Conceptualization:** Soya Kawabata, Noriaki Kurita, Takuya Nikaido, Nobuyuki Fujita, Seiji Ohtori.

**Formal analysis:** Soya Kawabata, Noriaki Kurita.

**Funding acquisition:** Takuya Nikaido.

**Investigation:** Soya Kawabata, Ryoji Tominaga, Yuji Endo.

**Methodology:** Soya Kawabata.

**Supervision:** Noriaki Kurita, Takuya Nikaido, Nobuyuki Fujita, Shin-ichi Konno, Seiji Ohtori.

**Writing – original draft:** Soya Kawabata, Nobuyuki Fujita.

**Writing – review & editing:** Soya Kawabata, Noriaki Kurita, Takuya Nikaido, Ryoji Tominaga, Yuji Endo, Nobuyuki Fujita, Shin-ichi Konno, Seiji Ohtori.

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
