## [Decision Letter · Decision Letter 0]

PONE-D-25-17481Association between lifestyle-related factors and low back pain: Evidence from a Japanese population–based studyPLOS ONE

Dear Dr. Fujita,

Thank you for submitting your manuscript to PLOS ONE. After careful consideration, we feel that it has merit but does not fully meet PLOS ONE’s publication criteria as it currently stands. Therefore, we invite you to submit a revised version of the manuscript that addresses the points raised during the review process.

We look forward to receiving your revised manuscript.

Kind regards,

Hiroshi Hashizume, M.D., Ph.D.

Academic Editor

PLOS ONE

Journal Requirements:

“This study was supported by a scholarship donation from Eli Lilly Japan K.K., and a research grant from the Japanese Society of Lumbar Spine Disorders. “

“This study was supported by a scholarship donation from Eli Lilly Japan K.K.”

4. In the online submission form, you indicated that “The data underlying this study cannot be shared publicly due to participant confidentiality and ethical restrictions. The participants did not provide consent for public data sharing, and the data include potentially identifying information. Access to the data can be requested from the corresponding author, pending approval from the Ethics Committee of Fukushima Medical University.”

6. One of the noted authors is a group or consortium “Clinical Research Committee of the Japanese Society of Lumbar Spine Disorders” In addition to naming the author group, please list the individual authors and affiliations within this group in the acknowledgments section of your manuscript. Please also indicate clearly a lead author for this group along with a contact email address.

**Additional Editor Comments:**

Two reviewers have evaluated your manuscript. The editor believes that Reviewer 2's comments will be useful in improving the quality of your paper. I look forward to your response.

Reviewers' comments:

Reviewer's Responses to Questions

**Comments to the Author**

1. Is the manuscript technically sound, and do the data support the conclusions?

Reviewer #1: Yes

Reviewer #2: Yes

2. Has the statistical analysis been performed appropriately and rigorously? 

Reviewer #1: Yes

Reviewer #2: Yes

3. Have the authors made all data underlying the findings in their manuscript fully available?

Reviewer #1: Yes

Reviewer #2: Yes

4. Is the manuscript presented in an intelligible fashion and written in standard English?

Reviewer #1: Yes

Reviewer #2: Yes

5. Review Comments to the Author

Reviewer #1: Thank you for submitting great article that this cross-sectional study investigated the relationship between lifestyle-related factors (LRFs) and low back pain (LBP) in a Japanese population. Using data from a nationwide survey of 2,188 randomly selected Japanese adults, the researchers examined how various LRFs relate to the presence, severity, and chronicity of LBP.

The study found that several LRFs were significantly associated with current LBP: higher body mass index (BMI), alcohol consumption, smoking, and dyslipidemia. When analyzing pain severity, smoking, lack of exercise, and dyslipidemia were associated with moderate/severe pain. For chronic LBP (lasting more than 3 months), smoking was the only significant LRF identified. The researchers concluded that LRFs play a crucial role in LBP, with smoking and dyslipidemia being particularly notable risk factors. Smoking was the only factor associated with all three aspects: presence, severity, and chronicity of LBP. I have some comments for publishment.

Q1. Please provide more specific details about how participants were recruited. Was the invitation sent by mail using the residential registry, or was the study advertised through email or telephone? The recruitment method should be clearly stated as it could potentially introduce selection bias in your study population.

Q2. Regarding the comorbidities mentioned (dyslipidemia, diabetes, and hypertension), please clarify whether participants who were receiving treatment, such as medication, were included in these groups. This information is important for understanding the composition of your study population and could impact the interpretation of your results.

Q3. The rationale for investigating self-image regarding body shape (such as "Obese") is not explained in the Introduction section. Please provide justification for including this variable in your study. The reader needs to understand why self-perception of body image was considered relevant to low back pain, and how this connects to your theoretical framework or previous research findings.

Q4. Please provide a more in-depth discussion about why dyslipidemia was associated with pain severity but not with chronic LBP. The authors suggest that this relationship might be due to the association between dyslipidemia and disc degeneration. However, it is well established that disc degeneration increases with age, and this aspect was not specifically examined in your study.

I recommend developing a more nuanced interpretation of these findings. For instance, could there be different pathophysiological mechanisms at play in acute versus chronic pain? Might dyslipidemia affect nociceptive processing or inflammatory pathways that influence pain severity without necessarily contributing to pain chronicity? Alternatively, could there be confounding factors or mediating variables that explain this seemingly paradoxical relationship? A more comprehensive discussion of these possibilities would strengthen the paper.

Q5. I notice that despite finding a significant association between alcohol consumption and current LBP (OR=1.37, 95% CI: 1.04-1.80, P=0.025), there is no discussion of this relationship in your paper. Please add commentary on this finding in the Discussion section.

Consider addressing potential mechanisms through which alcohol consumption might influence LBP, whether there are dose-dependent effects, how your findings compare with previous literature on this topic, and any clinical implications. This would provide a more complete interpretation of all significant results identified in your study.

Reviewer #2: The authors reported that multiple lifestyle-related factors (LRFs) are associated with the prevalence of low back pain (LBP) in a cross-sectional study of 2,188 subjects. Among the LRFs, dyslipidemia was associated with the presence and severity of LBP, whereas smoking was associated with both its severity and chronicity.

Both dyslipidemia and smoking are well-known risk factors for low back pain (LBP), as supported by accumulating evidence from previous reports. Dyslipidemia has been reported as a contributing factor to disc degeneration; however, its association with lumbar spinal epidural lipomatosis and canal stenosis may also correlate with the presence and severity of LBP (Fujita. Spine Surg Relat Res 2021; 5(2): 61-67).

I hope this manuscript encourages all spine surgeons to focus not only on surgical treatment but also on improving lifestyle-related factors.

I am pleased to inform the authors that this manuscript is worthy of publication in PLOS ONE.

6. PLOS authors have the option to publish the peer review history of their article (what does this mean?). If published, this will include your full peer review and any attached files.

Reviewer #1: **Yes: **Masatoshi Teraguchi

Reviewer #2: No

---

## [Author Response · Author response to Decision Letter 1]

29 May 2025

Response to Reviewers

We sincerely thank the Academic Editor and the reviewers for their thoughtful evaluation of our manuscript. Please find enclosed the revised version of our manuscript along with a detailed response to the reviewers’ comments. The feedback provided was highly valuable in guiding our revisions. Changes made in the manuscript are highlighted in yellow. We believe that these revisions have substantially improved the quality of the manuscript.

Academic editor

Response)

We have revised the manuscript to ensure that it adheres to PLOS ONE’s style requirements, including formatting and file naming conventions.

“This study was supported by a scholarship donation from Eli Lilly Japan K.K., and a research grant from the Japanese Society of Lumbar Spine Disorders. “

Response)

We have added the following statement to clarify the role of the funders, as requested:

This sentence has been included in the cover letter accordingly.

“This study was supported by a scholarship donation from Eli Lilly Japan K.K.”

Response)

We appreciate the helpful guidance. We have added the following statement to both the cover letter and the manuscript, as requested:

"This does not alter our adherence to PLOS ONE policies on sharing data and materials."

4. In the online submission form, you indicated that “The data underlying this study cannot be shared publicly due to participant confidentiality and ethical restrictions. The participants did not provide consent for public data sharing, and the data include potentially identifying information. Access to the data can be requested from the corresponding author, pending approval from the Ethics Committee of Fukushima Medical University.”

Response)

We apologize for the oversight. The full dataset cannot be made publicly available due to ethical restrictions and participant confidentiality, as informed consent for open data sharing was not obtained and the data contain potentially identifying information. Interested researchers may request access to the data by contacting the Clinical Research Committee of the Japanese Society of Lumbar Spine Disorders via Dr. Takuya Nikaido (Associate Professor, Department of Orthopaedic Surgery, Fukushima Medical University School of Medicine; Email: nikataku@gmail.com), pending approval from the Ethics Committee of Fukushima Medical University. We hope this approach complies with the journal’s data sharing policy, and we appreciate your understanding regarding the ethical considerations involved.

Response)

The corresponding author already has an ORCID iD. We will complete the validation process in Editorial Manager as requested.

6. One of the noted authors is a group or consortium “Clinical Research Committee of the Japanese Society of Lumbar Spine Disorders” In addition to naming the author group, please list the individual authors and affiliations within this group in the acknowledgments section of your manuscript. Please also indicate clearly a lead author for this group along with a contact email address.

Response)

We appreciate the helpful guidance. We have decided to list the "Clinical Research Committee of the Japanese Society of Lumbar Spine Disorders" as the affiliated organization rather than listing the individual authors.

Response)

We have added several relevant references and corrected previously inappropriate or inaccurate citations in the reference list. All changes have been made to ensure accuracy and consistency with PLOS ONE’s formatting and citation guidelines.

Reviewer #1:

Q1. Please provide more specific details about how participants were recruited. Was the invitation sent by mail using the residential registry, or was the study advertised through email or telephone? The recruitment method should be clearly stated as it could potentially introduce selection bias in your study population.

Response)

We appreciate the reviewer’s valuable feedback. In response to the reviewer’s comment, we deleted the sentence “Surveyors visited the participants’ residences to distribute and collect the survey forms” due to insufficient explanation, and created a new sentence to clarify the procedure in more detail.

Page 6, lines 94 to 98:

Before the survey, a letter requesting participation was sent in advance to the selected individuals. Subsequently, trained surveyors visited their homes and hand-delivered the self-administered survey forms enclosed in detention envelopes. Upon retrieval, the surveyors confirmed directly with the participants or their household members that the forms had been completed by the participants themselves.

Page 6, lines 101:

Participants were rewarded with a gift card for completing the survey.

Q2. Regarding the comorbidities mentioned (dyslipidemia, diabetes, and hypertension), please clarify whether participants who were receiving treatment, such as medication, were included in these groups. This information is important for understanding the composition of your study population and could impact the interpretation of your results.

Response)

We appreciate the reviewer’s insightful comment. As the information regarding comorbidities was obtained through a self-administered questionnaire, we were only able to assess the presence or absence of conditions such as dyslipidemia, diabetes, and hypertension. Unfortunately, we did not collect data on whether participants were receiving treatment (e.g., medication) for these conditions. We acknowledge that the treatment status may influence the interpretation of our results, and we have clarified this limitation in the manuscript accordingly.

Page 7, lines 122:

; however, treatment status for these comorbidities was not assessed

Q3. The rationale for investigating self-image regarding body shape (such as "Obese") is not explained in the Introduction section. Please provide justification for including this variable in your study. The reader needs to understand why self-perception of body image was considered relevant to low back pain, and how this connects to your theoretical framework or previous research findings.

Response)

We appreciate the reviewer’s thoughtful comment.

In response, we have added the following sentence and a relevant reference to the manuscript to clarify the rationale for including self-image regarding body shape. This addition reflects previous findings that self-perceived body shape may impact health behaviors and comorbidities, and supports its relevance in the context of LBP.

Page 4, lines 65 to 68:

“Beyond these physical and clinical indicators, self-image regarding body shape—a form of subjective health perception—has also gained attention as a psychosocial factor influencing lifestyle-related diseases and overall health status [11].”

Page 21, lines 367 to Page 22, lines 370:

Ramos-Vera C, Quispe-Callo G, Basauri-Delgado M, Calizaya-Milla YE, Casas-Gálvez C, Gálvez-Díaz NDC, et al.The mediating role of healthy behaviors and self-perceived health in the relationship between eating behaviors and comorbidity in adults. Arch Public Health. 2024;82(1):203. doi: 10.1186/s13690-024-01435-w

Q4. Please provide a more in-depth discussion about why dyslipidemia was associated with pain severity but not with chronic LBP. The authors suggest that this relationship might be due to the association between dyslipidemia and disc degeneration. However, it is well established that disc degeneration increases with age, and this aspect was not specifically examined in your study.

I recommend developing a more nuanced interpretation of these findings. For instance, could there be different pathophysiological mechanisms at play in acute versus chronic pain? Might dyslipidemia affect nociceptive processing or inflammatory pathways that influence pain severity without necessarily contributing to pain chronicity? Alternatively, could there be confounding factors or mediating variables that explain this seemingly paradoxical relationship? A more comprehensive discussion of these possibilities would strengthen the paper.

Response)

We appreciate the reviewer’s insightful comment regarding the differential association of dyslipidemia with pain severity and chronicity. In response, we have expanded the Discussion section to include a possible explanation based on existing literature. Specifically, we now note that this discrepancy may be due to confounding factors such as obesity and physical inactivity, which are closely related to both dyslipidemia and chronic LBP. For example, a large cohort study involving more than 18,000 adults found that low HDL-C and high triglyceride levels initially predicted an increased risk of chronic LBP, but these associations disappeared after adjusting for BMI and lifestyle variables. We have cited this study and suggested that the lack of a significant association between dyslipidemia and chronic LBP in our analysis may reflect the influence of such unmeasured confounders or mediators. We believe this addition provides a more nuanced interpretation of our findings.

Page 16, lines 254 to 259:

These discrepancies may be explained by confounding factors such as obesity and physical inactivity, which are often associated with both dyslipidemia and chronic LBP. For example, in a large cohort study involving over 18,000 adults, low HDL-C and high triglyceride levels were initially associated with an increased risk of chronic LBP. However, these associations disappeared after adjusting for body mass index and lifestyle variables [8]. This suggests that the link between dyslipidemia and chronic pain may be mediated by adiposity rather than a direct causal effect. In our study, the absence of a significant association between dyslipidemia and chronic LBP may similarly reflect the influence of unmeasured confounders or mediating variables [8].

Q5. I notice that despite finding a significant association between alcohol consumption and current LBP (OR=1.37, 95% CI: 1.04-1.80, P=0.025), there is no discussion of this relationship in your paper. Please add commentary on this finding in the Discussion section.

Consider addressing potential mechanisms through which alcohol consumption might influence LBP, whether there are dose-dependent effects, how your findings compare with previous literature on this topic, and any clinical implications. This would provide a more complete interpretation of all significant results identified in your study.

Response)

We appreciate the reviewer’s thoughtful comment regarding the lack of discussion on the association between alcohol consumption and current LBP. In response, we have added the following paragraph to the Discussion section. We believe this addition addresses the comment by providing context from the literature and acknowledging the clinical implications of our findings.

Page 17, lines 277 to 284:

Furthermore, this study identified alcohol consumption as an independent risk factor for current LBP. According to a systematic review by Ferreira et al., alcohol consumption is slightly associated with LBP; however, no dose-response relationship was identified [19]. Additionally, Vaajala et al. reported that excessive alcohol intake during adolescence increases the risk of developing LBP in adulthood [18]. Based on these findings, it is advisable to avoid excessive alcohol consumption from the perspective of LBP prevention.

Page 23, lines 394 to 396:

Ferreira PH, Pinheiro MB, Machado GC, Ferreira ML. Is alcohol intake associated with low back pain? A systematic review of observational studies. Man Ther. 2013;18(3):183–190. doi:10.1016/j.math.2012.11.008

Reviewer #2:

The authors reported that multiple lifestyle-related factors (LRFs) are associated with the prevalence of low back pain (LBP) in a cross-sectional study of 2,188 subjects. Among the LRFs, dyslipidemia was associated with the presence and severity of LBP, whereas smoking was associated with both its severity and chronicity.

Both dyslipidemia and smoking are well-known risk factors for low back pain (LBP), as supported by accumulating evidence from previous reports. Dyslipidemia has been reported as a contributing factor to disc degeneration; however, its association with lumbar spinal epidural lipomatosis and canal stenosis may also correlate with the presence and severity of LBP (Fujita. Spine Surg Relat Res 2021; 5(2): 61-67).

I hope this manuscript encourages all spine surgeons to focus not only on surgical treatment but also on improving lifestyle-related factors.

I am pleased to inform the authors that this manuscript is worthy of publication in PLOS ONE.

Response)

We appreciate the reviewer’s positive evaluation and insightful comments. In response to the suggestion, we have added a sentence to the Discussion to elaborate on the potential involvement of dyslipidemia in lumbar spinal epidural lipomatosis and canal stenosis, as described by Fujita et al. (Spine Surg Relat Res 2021). This addition supports the broader pathophysiological link between dyslipidemia and LBP severity and presence, beyond disc degeneration alone. We also appreciate the r

---

## [Decision Letter · Decision Letter 1]

Association between lifestyle-related factors and low back pain: Evidence from a Japanese population–based study

PONE-D-25-17481R1

Dear Dr. Fujita,

We’re pleased to inform you that your manuscript has been judged scientifically suitable for publication and will be formally accepted for publication once it meets all outstanding technical requirements.

Kind regards,

Hiroshi Hashizume, M.D., Ph.D.

Academic Editor

PLOS ONE

Additional Editor Comments (optional):

The authors completed the revision very well according to the reviewers’ instructions. Congratulations.

Reviewers' comments:

Reviewer's Responses to Questions

**Comments to the Author**

1. If the authors have adequately addressed your comments raised in a previous round of review and you feel that this manuscript is now acceptable for publication, you may indicate that here to bypass the “Comments to the Author” section, enter your conflict of interest statement in the “Confidential to Editor” section, and submit your "Accept" recommendation.

Reviewer #1: All comments have been addressed

Reviewer #2: All comments have been addressed

2. Is the manuscript technically sound, and do the data support the conclusions?

Reviewer #1: Yes

Reviewer #2: Yes

3. Has the statistical analysis been performed appropriately and rigorously? 

Reviewer #1: Yes

Reviewer #2: Yes

4. Have the authors made all data underlying the findings in their manuscript fully available?

Reviewer #1: Yes

Reviewer #2: Yes

5. Is the manuscript presented in an intelligible fashion and written in standard English?

Reviewer #1: Yes

Reviewer #2: Yes

6. Review Comments to the Author

Reviewer #1: After careful review of the revised manuscript, I am pleased to confirm that the authors have successfully addressed all previously identified concerns. The modifications made to the paper have significantly strengthened its scholarly contribution, and I believe the revised work is now methodologically sound, well-structured, and presents findings that will be of considerable value to the academic community. Therefore, I consider this revised paper to be entirely appropriate and highly worthy of publication in your esteemed journal.

Reviewer #2: The authors have appropriately revised their manuscript to a level suitable for publication in PLOS ONE. I am pleased to recommend acceptance of this manuscript for publication in the journal.

7. PLOS authors have the option to publish the peer review history of their article (what does this mean?). If published, this will include your full peer review and any attached files.

Reviewer #1: No

Reviewer #2: No

---

## [Editor Report · Acceptance letter]

PONE-D-25-17481R1

PLOS ONE

Dear Dr. Fujita,

I'm pleased to inform you that your manuscript has been deemed suitable for publication in PLOS ONE. Congratulations! Your manuscript is now being handed over to our production team.

Kind regards,

on behalf of

Dr Hiroshi Hashizume

Academic Editor

PLOS ONE